# The 3q Oncogene *SEC62* Predicts Response to Neoadjuvant Chemotherapy and Regulates Tumor Cell Migration in Triple Negative Breast Cancer

**DOI:** 10.3390/ijms24119576

**Published:** 2023-05-31

**Authors:** Julia C. Radosa, Mariz Kasoha, Merle Doerk, Annika Cullmann, Askin C. Kaya, Maximilian Linxweiler, Marc P. Radosa, Zoltan Takacs, Andrea Tirincsi, Sven Lang, Martin Jung, Julian Puppe, Barbara Linxweiler, Mathias Wagner, Rainer M. Bohle, Erich-Franz Solomayer, Julia S. M. Zimmermann

**Affiliations:** 1Department of Gynecology, Obstetrics and Reproductive Medicine, Saarland University Hospital, D-66421 Homburg, Germanyjulia.zimmermann@uks.eu (J.S.M.Z.); 2Department of Otorhinolaryngology, Head and Neck Surgery, Saarland University Hospital, D-66421 Homburg, Germany; 3Department of Gynecology and Obstetrics, Klinikum Bremen Nord, D-28755 Bremen, Germany; 4Medical Biochemistry and Molecular Biology, Saarland University, D-66421 Homburg, Germany; 5Department of Gynecology, University Hospital Cologne, D-50931 Cologne, Germany; 6Department of Pathology, Saarland University Hospital, D-66421 Homburg, Germany

**Keywords:** triple-negative breast cancer, neoadjuvant chemotherapy, oncogene *SEC62*, tumor pathogenesis, translational research, tumor cell migration

## Abstract

In the absence of targeted treatment options, neoadjuvant chemotherapy (NACT) is applied widely for triple-negative breast cancer (TNBC). Response to NACT is an important parameter predictive of oncological outcomes (progression-free and overall survival). An approach to the evaluation of predictive markers enabling therapy individualization is the identification of tumor driver genetic mutations. This study was conducted to investigate the role of *SEC62*, harbored at 3q26 and identified as a driver of breast cancer pathogenesis, in TNBC. We analyzed *SEC62* expression in The Cancer Genome Atlas database, and immunohistologically investigated *SEC62* expression in pre- and post-NACT tissue samples from 64 patients with TNBC treated at the Department of Gynecology and Obstetrics/Saarland University Hospital/Homburg between January 2010 and December 2018 and compared the effect of *SEC62* on tumor cell migration and proliferation in functional assays. *SEC62* expression dynamics correlated positively with the response to NACT (*p* ≤ 0.01) and oncological outcomes (*p* ≤ 0.01). *SEC62* expression stimulated tumor cell migration (*p* ≤ 0.01). The study findings indicate that *SEC62* is overexpressed in TNBC and serves as a predictive marker for the response to NACT, a prognostic marker for oncological outcomes, and a migration-stimulating oncogene in TNBC.

## 1. Introduction

Triple-negative breast cancers (TNBC) are malignant tumors characterized by a lack of estrogen receptor (ER) and progesterone receptor (PR) expression and human epidermal growth factor receptor 2 (HER2)/neu overexpression [1,2]. TNBC accounts for an estimated 10 to 20% of breast cancers in women overall [1]. These cancers tend to be more aggressive and have a poorer prognosis compared with other breast cancer subtypes [2,3,4]. Given the lack of targeted treatment options, neoadjuvant chemotherapy (NACT) is most frequently applied [5]. The response to NACT is one of the most important parameters predictive of the overall oncological outcome, and markers predicting this response have been a main focus of interest in breast cancer research to date. One possible approach to the identification of predictive markers that would enable treatment individualization is the identification of tumor-specific (“driver”) genetic mutations. 3q26 amplifications are driver mutations known to play roles in the pathogenesis of various cancer entities, including prostate, head and neck, non-small cell lung, ovarian, and breast cancers [6,7,8,9]. Numerous studies have been conducted with the aim of identifying oncogene candidates; *PIK3CA*, *TP63*, *CLAPM1*, and *FXR1*, encoded on the long arm of chromosome 3, have shown no functional correlation with breast tumorigenesis [8,10,11]. *SKIL* and *SEC62* were identified as 3q26-resident tumor driver genes, as their overexpression contributes to the malignant transformation of lung, ovarian, and breast cancers [8]. *SEC62* encodes a transmembrane protein (Sec62) of the endoplasmic reticulum that participates in protein translocation across the endoplasmic reticulum membrane, endoplasmic reticulum–phagy to compensate cellular stress, and the maintenance of intracellular calcium homeostasis [12,13]. In vitro studies conducted with tissue samples and cancer cell lines have shown that *SEC62* is overexpressed in tumors relative to tumor-free tissue at the protein and mRNA levels in lung, prostate, head and neck, and cervical cancers [14,15,16,17]. Although the precise physiological function of Sec62 is not completely understood, this protein seems to enhance the stress tolerance, migration, and invasive potential of *SEC62*-overexpressing cells—the three hallmarks of cancer—resulting in high rates of lymph node metastasis and poorer overall prognoses [14,15,16,17,18,19].

Given the identification of *SEC62* as a potential driver of breast cancer development and its prognostic impact on other cancers [8], we further studied its role in the pathogenesis of breast cancer. In a pilot study, we found that Sec62 protein expression was greater in breast cancer tissue samples than in tumor-free tissue samples from the same patients and that this high expression correlated with distant metastasis and poor overall survival (OS) [19]. Additionally, we found that the *SEC62* expression level differed among breast cancer types and was highest in a small sample of patients with TNBC [19]. In this study, we further evaluated the function of *SEC62* in TNBC, focusing on its role in predicting the response to NACT and its function in tumor cell biology. We examined genomic *SEC62* expression using The Cancer Genome Atlas (TCGA) database and Sec62 protein levels in tissue samples from patients with TNBC before and after NACT completion and performed functional assays to assess the impact of *SEC62* on TNBC cell migration and proliferation.

## 2. Results

### 2.1. Frequency of SEC62 Alterations in TCGA Data

*SEC62* alterations, classified as mutations, amplifications, deep deletions, and multiple alterations, were detected in 10,967 (12.61%) of 86,962 analyzed cases of 32 tumor entities in the TCGA database. Breast cancer ranked 13th among the most altered tumors (3.5% alteration frequency), and the predominance of amplifications (88%) was found in breast cancer cases. The remaining 12% of *SEC62* alterations in the breast cancer cohort were mutations. Four other gynecological malignancies (ovarian, cervical, uterine, and endometrial cancers) were ranked among the top seven tumor entities with the highest frequencies of genomic *SEC62* alteration (Figure 1a). For all cancer entities recorded in TCGA, OS durations were significantly shorter for patients with than for those without *SEC62* alterations (*p* ≤ 0.01, Figure 1b).

### 2.2. Immunohistochemical Analysis of SEC62 Expression in TNBC Tissue Samples

Immunohistochemical (IHC) *SEC62* expressions were examined in pre- and post-NACT samples from 64 patients with primary TNBC; samples from four other patients were excluded due to insufficient slide quality and the lack of additional material for repeat staining. The patient and tumor characteristics are shown in Table 1.

In the analyzed tissue slides, we observed cytoplasmic Sec62 positivity in all TNBC cells and not in physiological breast tissue cells (Figure 2).

Immunoreactive scores [IRS; range, 0 (negative) to 12 (high)] reflected significantly greater median *SEC62* expression in core biopsy tissue specimens (prior to NACT) than in final specimens (post-NACT) [8 (range, 2–12) vs. 4 (range, 0–12); *p* ≤ 0.01, Figure 2, Table 2]. The median difference in the IRS between these specimen types was 4 (range, −6–12) points (Table 2).

The median regression grade, according to Sinn et al. [20], was higher for tumors with core biopsy IRSs > 8 than for those with IRSs ≤ 8 [3 (range, 1–4) vs. 2 (range, 0–4); *p* ≤ 0.01]. Final specimen IRS correlated inversely with the regression grade [1 (range, 0–2) vs. 4 (range, 0–4); *p* ≤ 0.01]. For IRS differences between core biopsy and final specimens, we found a cutoff of ≥ 6 to be relevant for the identification of cases with higher median regression grades [4 (range, 2–4) vs. 1 (range, 0–4); *p* ≤ 0.01, Table 3]. Linear regression analysis [analysis of variance (ANOVA)] showed that the regression grade was affected directly by the cT stage [odds ratio (OR) –0.35; 95% confidence interval (CI) –0.66 to –0.04; *p* = 0.03] and a difference ≥6 between the core biopsy and final specimen IRSs (OR 1.75; 95% CI, 1.16–2.34; *p* ≤ 0.01, Table 4).

### 2.3. Correlation of Patient Survival with SEC62 Expression

When analyzing correlations between *SEC62* expression and progression-free survival (PFS), no correlation between the mean PFS and an IRS cutoff of 8 on core biopsy (77.8 months (95% CI 62.0–93.6), 61.9% vs. 60.1 months (95% CI 40.5–79.7), 60.0%; *p* = 0.69) or between PFS and an IRS cutoff of 8 on final specimen (78.1 months (95% CI 64.0–92.2), 63.0% vs. 50.2 months (95% CI 19.7–82.6), 44.4%; *p* = 0.23) was detected. However, we observed a mean PFS of 89.5 (95% CI 74.9–104.2) months (83.3%) in the group of patients with a difference of the Sec62 IRS ≥ 6 between pre- and post-treatment, compared with 60.7 (95% CI 43.5–77.9) months (47.4%) in the group of patients with a difference of the Sec62 IRS < 6 (*p* ≤ 0.01) (Figure 3a). For overall survival (OS), an IRS cutoff of 8 on core biopsy (82.7 months (95% CI 68.9–97.4), 64.3% vs. 108.0 months (95% CI 78.7–137.3), 75.0%; *p* = 0.51) or on the final specimen (106.1 months (95% CI 88.9–123.4), 70.4% vs. 56.1 months (95% CI 27.7–84.6), 44.4%; *p* = 0.08) did not correlate with the mean OS. Though, we detected a mean OS of 98.6 (95% CI 88.9–100.3) months (91.7%) in the group of patients with a difference of the Sec62 IRS ≥ 6 between pre- and post-treatment and a mean OS of 84.6 (95% CI 63.5–105.7) months (52.6%) in the group of patients with a difference of the Sec62 IRS < 6 between pre- and post-treatment (*p* ≤ 0.01) (Figure 3b).

### 2.4. Effect of SEC62 Expression on CAL-120 (TNBC) Cell Proliferation and Migratory Potential

After the transfection of CAL-120 cells with two different *SEC62*-targeting siRNAs, median Sec62 protein levels were reduced to 41.25% for the SEC62#2 siRNA (*p* ≤ 0.01) and to 42.75% for the SEC62-UTR siRNA (*p* ≤ 0.01) (Figure 4a,b), which did not affect TNBC cell proliferation after 96 h (Figure 4c,d). We observed a temporary, but not statistically significant effect of *SEC62*-silencing on cell proliferation after 45 h with median cell indices of 6.78 (range 6.70–7.15) for the control siRNA, 6.21 (range 6.01–6.85) for the SEC62#2 siRNA (*p* = 0.11), and 5.08 (range 5.03–5.64) for the SEC62-UTR siRNA (*p* = 0.11) (Figure 4c). Median cell indices after 96 h were 7.5 (range 7.4–7.6) for the control siRNA, 7.6 (range 7.5–7.7) for the SEC62#2 siRNA (*p* = 0.18), and 7.4 (range 7.3–7.5) for the SEC62-UTR siRNA (*p* = 0.18) (Figure 4c,d). In the migration assays, *SEC62* knock-down significantly reduced the cells’ migratory potential, with median cell migration arbitrary units of 263 (range 241–279) for the control siRNA, 178 (range 96–276) for the siSEC62#2 (*p* ≤ 0.01), and 72 (range 70–149) for the siSEC62-UTR (*p* ≤ 0.01) (Figure 5a,b).

The induction of further *SEC62* overexpression in CAL-120 cells via transfection with a *SEC62*-encoding expression plasmid did not alter cell proliferation or increase cell migration as compared with control plasmid-transfected cells (Figure 6a–d). The median cell indices were 6.77 (range 5.66–6.9) for the control plasmid and 6.62 (range 6.46–7.16) for the *SEC62*-encoding expression plasmid (*p* = 0.29). The median migrated cells in the control plasmid group were 162 (range 161–220) and 182 (range 173–285) for the *SEC62*-encoding expression plasmid (*p* = 0.11). We attribute this to the observation that there already is a high level of *SEC62* overexpression associated with TNBC (Figure 2). Notably, in previous experiments, *SEC62* overexpression stimulated the migratory potential and stress tolerance of otherwise poorly migrating cells, such as HEK293, HeLa, and FaDu cells [13,17].

## 3. Discussion

In this study, we found that the 3q-encoded oncogene *SEC62* is amplified in about 3% of all breast cancer cases and overexpressed in TNBC cells compared with physiological breast epithelial tissue. We determined that *SEC62* expression in TNBC is a predictor of the response to NACT with 4× Epirubicin 90 mg/m^2^ and Cyclophosphamid 600 mg/m^2^ i.v. q2w, followed by 12× Paclitaxel 80 mg/m^2^ and Carboplatin AUC 2 i.v. q1w and that the Sec62 level in TNBC cells affects their migration. Given these findings, *SEC62* may be identified as a tumor driver gene as a result of 3q amplification in TNBC and might serve as a potential therapeutic target in this context.

Our analyses of TCGA data, which showed *SEC62* alterations (mostly amplifications) in 32 tumor entities and shortened OS for patients with these alterations, support the potential role of *SEC62* as a key tumor driver gene in various cancer entities, including breast cancer. These findings are consistent with those of Hagerstrand et al. [8], who first reported that *SEC62* had an oncogenic effect through its enhancement of the migratory and invasive potential of tumor cells. Based on findings of functional assays and genomic database analyses, *SEC62* has been identified as a potential driver of ovarian, lung, prostate, and head and neck cancers, and high *SEC62* expression has been correlated with significantly shorter disease-free survival and overall survival [8,9,14,16,17].

We identified a potential predictive role of *SEC62* expression for the response to NACT with 4× Epirubicin 90 mg/m^2^ and Cyclophosphamid 600 mg/m^2^ i.v. q2w, followed by 12× Paclitaxel 80 mg/m^2^ and Carboplatin AUC 2 i.v. q1w, and thus overall prognosis, based on Sec62 IRS differences ≥6 between core biopsy and final specimens. This difference was the only factor influencing the response to NACT in a linear regression analysis, indicating that the reduction of *SEC62* expression during NACT affects the prognosis of TNBC. IRS differences <6 were associated with shortened PFS and OS for patients with TNBC. An interesting observation in this context regards the importance of Sec62 IRS dynamics during therapy and their role in predicting pathological complete response. As the degree of pathological regression translates to improved PFS and OS for patients with TNBC, these results are promising and might be used to further improve and tailor therapy regimens for these patients in the future. Concepts towards a more tailored and targeted therapy for patients with TNBC have recently involved aiming at the selection of patients in need of additional post-neoadjuvant treatment options [21]. Such selection is driven mainly by the degree of response to neoadjuvant treatment measured pathologically by the regression grade according to Sinn et al. [20], although a shift towards the implementation of molecular markers, such as the *BRCA* mutation status, has recently evolved [2,21,22]. Thus, *SEC62* expression may serve as a promising tool in order to personalize and tailor therapies in the future.

To the best of our knowledge, this study is the first to assess the ability of *SEC62* expression to predict the response to NACT and the correlation of *SEC62* dynamics with PFS and OS in breast cancer. However, our results are in line with the preliminary finding of Takacs and colleagues [19] that *SEC62* plays a prognostic role in breast cancer. They evaluated the IHC Sec62 staining intensity in tissue samples from 53 patients using an IRS cut-off of 8 and determined that OS was worse in patients with *SEC62* overexpression, independent of the breast cancer type [19]. Based on microarrays of 102 colorectal cancer tissue samples and in vivo functional analyses, Liu et al. [23] determined that *SEC62* promoted chemoresistance and that its depletion sensitized colorectal cancer cells to chemotherapeutic drugs. We found that pre- and post-NACT IRS differences ≥ 6 correlated positively with an improved response to NACT with 4× Epirubicin 90 mg/m^2^ and Cyclophosphamid 600 mg/m^2^ i.v. q2w, followed by 12× Paclitaxel 80 mg/m^2^ and Carboplatin AUC 2 i.v. q1w, but we cannot comment on the potential impact of *SEC62* silencing on the therapeutic response because we did not perform the relevant functional assays. The differences in the findings may also be due to differences in the cancer type and chemotherapeutic regimen.

Correlations of IHC-detected *SEC62* overexpression with poor clinical outcomes (i.e., reduced PFS and OS) have been reported for ovarian, non-small cell lung, head and neck, hepatocellular, and prostate cancers, with sample sizes ranging from 22 to 167 patients [9,14,16,24,25]. In contrast, we found that *SEC62* dynamics, rather than overexpression, correlated with shortened PFS and OS in patients with TNBC. The dynamics of *SEC62* expression were not examined in the previous study [9,14,16,24,25].

Tumor cells are known to have marked migratory and invasive potential [18]. We found that transient Sec62 depletion did not affect the proliferation of CAL-120 cells after 96 h, but that *SEC62* silencing significantly inhibited cell migration. These results are consistent with previous reports of similar effects of *SEC62* gene silencing in cervical and head and neck cancer cell lines [15,26]. Linxweiler et al. [15] and Bochen et al. [26] assessed the migration and proliferation of HeLa and UM-SCC1 cells, respectively, using functional assays as we did, and found that *SEC62* gene silencing inhibited cell migration, but not proliferation. However, these findings do not explain the molecular role of *SEC62* in cell migration. *SEC62* encodes the transmembrane protein Sec62 of the endoplasmic reticulum, which appears to participate in the intracellular transport of proteins, endoplasmic reticulum–phagy, and cellular calcium homeostasis; the observed effect on cell migration may be due to the intracellular transport of involved proteins, but the exact physiological functions of Sec62 need to be studied further [27,28,29]. The inhibition of cell migration by *SEC62* silencing is an interesting potential target for new treatment options since the effect of *SEC62* silencing on the migratory potential of various cancer cells can be mimicked by treatment with trifluoperazine (TFP), an antipsychotic drug used in the treatment of schizophrenia [13,30]. Moreover, phase-II clinical trials assessing the use of the prodrug mipsagargin/G202 in the treatment of prostate cancer have been initiated. This pharmacologically distinct approach mimics *SEC62* silencing, as the drug directly targets sarcoplasmic/endoplasmic reticulum calcium ATPase, and thereby cellular calcium homeostasis [12,31]. Further cell culture and animal studies are needed to assess these treatment options.

The limitations of the present study include the lack of a statistical assessment of interobserver variability, a known source of error, in the IHC evaluations. We sought to account for such possible bias by using four independent observers. In addition, the study was limited to TNBC, the breast cancer type with the greatest *SEC62* expression [19]; the role of *SEC62* in other types of breast cancer needs to be evaluated further. Apart from that, another limitation of the present study was the fact that we could only comment on the response to NACT with the standard protocol 4× Epirubicin 90 mg/m^2^ and Cyclophosphamid 600 mg/m^2^ i.v. q2w, followed by 12× Paclitaxel 80 mg/m^2^ and Carboplatin AUC 2 i.v. q1w and did not investigate other chemotherapeutic regimens. A possible bias in the observed effect of *SEC62* silencing on cell migration might be the *SEC62-*mediated temporary effect on cell proliferation and hence an indirect effect due to reduced cellular fitness. These findings are supported by CRISPR screen analyses (https://depmap.org/portal/gene/SEC62?tab=dependency, accessed on 20 May 2023), which demonstrate *SEC62* to be relevant for CAL-120 cell proliferation and survival outcomes. However, the strong decrease in cell migration in contrast to the temporary effect on cell proliferation, hints at an independent effect of *SEC62* on cell migration.

## 4. Materials and Methods

### 4.1. Analysis of The Cancer Genome Atlas (TCGA) Data

An analysis of The Cancer Genome Atlas (TCGA) dataset regarding *SEC62* alterations, defined as mutations, amplifications, deep deletions, or multiple alterations, in different tumor entities was performed, using publicly available sequencing data from the National Cancer Institute’s GDC Data Portal [9]. The analysis was performed in February 2023 and included a total of 86,962 cases representing 68 primary tumor sites.

### 4.2. Patients and Tissue Samples

All eligible patients who received NACT with the standard protocol 4× Epirubicin 90 mg/m^2^ and Cyclophosphamid 600 mg/m^2^ i.v. q2w, followed by 12× Paclitaxel 80 mg/m^2^ and Carboplatin AUC 2 i.v. q1w for primary TNBC at the Department of Gynecology and Obstetrics, Saarland University Hospital, Homburg, Germany, between January 2010 and December 2018 were enrolled in this retrospective study. The scientific use of the patients’ tissue and clinical data as part of an oncological institutional biobank project and the study protocol were approved by the hospital’s institutional review board. The inclusion criteria were the availability of formalin-fixed paraffin-embedded (FFPE) tissue samples and complete clinical parameters and follow-up until at least 1 January 2019. TNBC was identified by immunohistological diagnostics during routine pathological examinations based on the absence of ER and PR expression (IHC staining of <1% of tumor cells) and lack of Her2 overexpression [IHC score of 0 or 1, or IHC score of 2 with no Her2 amplification (ratio < 2.0) on fluorescence in situ hybridization] [32]. Patient and tumor characteristics were obtained via clinical chart review.

### 4.3. IHC Analysis of Sec62

Representative FFPE blocks of primary tumor specimens and histologically tumor-free breast tissue were examined. A pathologist reconfirmed the presence of tumor tissue in all hematoxylin-stained tumor tissue samples. Pretreatment core breast biopsy tissue specimens and definitive pathological specimens obtained during breast surgery after NACT completion were subjected to IHC analysis. The first three 10 µm sections were discarded, and 3 µm sections were cut using a rotary microtome (RM 2235; Leica Microsystems, Wetzlar, Germany), transferred to Superfrost Ultra Plus Microscope slides (Menzel-Gläser, Braunschweig, Germany) and dried overnight in an incubator at 37 °C. After deparaffinization, heat-induced epitope retrieval was performed in retrieval solution (Dako S1699; Agilent Technologies, Santa Clara, CA, USA), and nonspecific protein-binding sites were blocked by incubation in a 3% bovine serum albumin (BSA)/phosphate-buffered saline (PBS) solution (Sigma-Aldrich Chemie GmbH, Taufkirchen, Germany) for 30 min at room temperature. Subsequently, the samples were incubated with a 1:800 solution (diluted in 1% BSA/PBS) of a specific affinity-purified polyclonal rabbit antipeptide antibody directed against the C terminus of human *SEC62* for 1 h at room temperature [9]. The house-made rabbit antibody was directed against the COOH terminal undecapeptide of the human Sec62 protein and an aminoterminal cysteine (peptide sequence in single letter code: CGETPKSSHEKS). Each staining series included positive and negative (without primary antibody) controls. Staining was visualized using the Dako real detection system (Agilent Technologies) according to the manufacturer’s instructions, and the slides were counterstained with hematoxylin (Dako Agilent Technologies, Glostrup, Denmark). Notably, commercial anti-Sec62 antibodies were described as a suitable alternative [23].

Four independent examiners (one pathologist and three gynecologists) with broad experience in IHC evaluation rated *SEC62* immunoreactivity using the IRS, a well-established and unbiased semiquantitative validation system for IHC assessment in breast cancer [33]. Staining intensity was classified as absent (0), weak (1), intermediate (2), and strong (3). The percentage of stained cells was classified as none (0), <10% (1), 10–50% (2), 51–80% (3), and >80% (4). The product of the scores for staining intensity and a number of stained cells are defined as the immunoreactive score (IRS) ranging from 0 (negative) to 12 (high). For the assessment of the Sec62 protein content of the tumor cells, we modified the scoring system and rated Sec62 “low” for a score of 0–8, and “high” for 9–12 as described in earlier studies [9,19]. *SEC62* expression in the two tumor tissue types was compared.

Based on our preliminary work and statistical considerations, IRS cutoffs of 8 and 6 were applied in further calculations [9,19]. The correlations of these cutoffs with the response to NACT and PFS and OS were evaluated. Treatment response was classified according to histologically observed regressive changes using the well-established semiquantitative system of Sinn et al. [20] as absent (0), resorption and tumor sclerosis (1), minimal (<0.5 cm) residual invasive tumor (2), residual noninvasive tumor only (3), and no tumor detectable (4). Additionally, *SEC62* expression in tumor and histologically tumor-free tissues from the same patients was compared.

### 4.4. Cell Biological Experiments

We compared the migratory potential of CAL-120 cells, which were transfected with control siRNA or one of two different *SEC62*-targeting siRNAs in the BD Falcon FluoroBlok system (BD, Franklin Lakes, NJ, USA). In addition, proliferation rates of the siRNA-treated cells were analyzed in real-time in the xCELLigence SP system (Roche Diagnostics GmbH, Mannheim, Germany). These siRNAs were previously established for various other cell types and were shown to cause efficient transient Sec62 depletion with little effect on cell proliferation during 96 h of cell growth [17]. Because of this latter observation by different laboratories, and as we wanted to be able to compare our results with those found for other tumor cells, we applied transient knock-down with siRNAs rather than CRISPR/Cas9 mediated knock-out strategies. 

### 4.5. Cell Culture and Transfection

CAL-120 cells, derived from a patient with invasive TNBC, were obtained from the German Collection of Microorganisms and Cell Cultures GmbH/DSMZ (ACC 459). They were cultured in Dulbecco’s modified Eagle medium (Gibco Invitrogen, Karlsruhe, Germany) containing 10% fetal bovine serum (FBS; Biochrom, Berlin, Germany) and 1% penicillin/streptomycin (PAA Laboratories GmbH, Pasching, Austria) in a humidified environment with 5% CO_2_ at 37 °C.

For gene silencing, 5.2 × 10^5^ CAL-120 cells were seeded onto 6 cm dishes and transfected with *SEC62*-siRNA (GGCUGUGGCCAAGUAUCUUtt; Ambion, Austin, TX, USA), siRNA directed against the 3′ untranslated region of *SEC62* (CGUAAAGUGUAUUCUGUACtt; Ambion), or control siRNA (AllStars; Qiagen, Hilden, Germany) using HiPerFect transfection reagent (Qiagen) according to the manufacturer’s instructions. After 24 h, the media were changed and the cells were transfected again for additional 24 h.

For the examination of additional *SEC62* overexpression, 5.2 × 10^5^ CAL-120 cells were seeded onto 6 cm dishes. After 24 h, the cells were transfected with the IRES-GFP *SEC62* plasmid or negative control plasmid using FuGENE transfection reagent (Promega, Madison, WI, USA) according to the manufacturer’s instructions, with pcDNA3 served as the parent for all plasmids.

### 4.6. Real-Time Cell Proliferation Analysis

The xCELLigence SP system (Roche Diagnostics GmbH, Mannheim, Germany) was used for the real-time analysis of cell proliferation. This system measures relative changes in impedance in specific 96-well plates with microelectrodes covering the well bottoms (E-plates; Roche Diagnostics GmbH), recorded as cell indices, a dimensionless parameter. CAL-120 cells (1 × 10^4^) were transfected with siRNA or plasmids and seeded into the 96-well plates according to the manufacturer’s instructions (at 24 h after the second transfection and 24 h after plasmid transfection, respectively). Cell proliferation was monitored for 120 h, and its rate was compared among groups using the RTCA 2.0 software (Roche Diagnostics GmbH).

### 4.7. Migration Potential Analysis

Cell migration was analyzed and compared using the Falcon FluoroBlok system (BD, Franklin Lakes, NJ, USA) with 8 μm pore inserts for 24-well plates. CAL-120 cells (5 × 10^4^) transfected with siRNA or plasmids were loaded into the inserts in a normal medium containing 1% FBS. The inserts were then placed in the wells of a 24-well plate in a medium containing 10% FBS as a chemoattractant for migration or no FBS (negative control). After 48 h, the cells were fixed with methanol, the nuclei were counterstained with DAPI, and three representative images of each insert were obtained using a ten-fold objective magnification. The number of migrated cells was determined and compared among groups using a bottom-reading fluorescence microscope and the NIS-Elements AR software (version 3.2; Nikon, Tokyo, Japan).All cell proliferation and migration experiments were repeated three times, and every cell population was analyzed in triplicate in each experiment, at a minimum.

### 4.8. Western Blot Analyses

Cells (1 × 10^6^) were lysed in a lysis buffer [distilled water, 10 mM NaCl, 10 mM Tris(hydroxymethyl)aminomethane, 3 mM MgCl_2_, and 5% NP-40], and the proteins were resolved by sodium dodecyl sulphate-polyacrylamide gel electrophoresis and identified by blotting and subsequent immunostaining. Canine pancreatic rough microsomes (RM) were analyzed in parallel and allowed the identification of the Sec62 protein in the human cell extracts [13,14,17,24]. We used the above-described affinity-purified polyclonal rabbit antipeptide antibody directed against the C terminus of human Sec62 (at a dilution of 1:500) and a monoclonal mouse antibody directed against the N terminus of human β-actin (at a dilution of 1:10,000) (Sigma-Aldrich, St. Louis, MO, USA). The secondary antibodies used were ECL Plex goat anti-rabbit Cy5 and anti-mouse Cy3 conjugates (GE Healthcare, Munich, Germany) (at dilutions of 1:1000 and 1:10,000, respectively). The blots were imaged using the Typhoon-Trio system and Image Quant TL software (version 7.0; GE Healthcare). The Sec62 and β-actin levels were quantified (in arbitrary signal units), and the former was normalized against the latter.

### 4.9. Statistical Analysis

The statistical analyses were performed using SPSS software (version 27; IBM Corporation, Armonk, NY, USA). Qualitative and quantitative data are presented as absolute and relative frequencies and medians and ranges, respectively. *SEC62* expression levels and their correlation with the response to NACT were compared using the Wilcoxon signed rank, Mann–Whitney, and Pearson chi-squared tests. Linear regression analysis (ANOVA) was performed to identify factors associated with the regression grade. Covariates for this analysis were selected based on the results of univariate analyses and clinical relevance. The Kaplan–Meier method was used for the univariate analysis of PFS and OS durations (in months). Survival curves were compared using the log-rank test. Cell proliferation and migration were analyzed using the D’Agostino and Person normality test and Wilcoxon signed rank test, Mann–Whitney test, and two-sided unpaired *t* test. Two-tailed *p* values < 0.05 were considered to be significant. 

## 5. Conclusions

This study showed that *SEC62* expression was increased in TNBC and predicted the response to NACT with 4× Epirubicin 90 mg/m^2^ and Cyclophosphamid 600 mg/m^2^ i.v. q2w, followed by 12× Paclitaxel 80 mg/m^2^ and Carboplatin AUC 2 i.v. q1w. This study also demonstrated that the dynamics of this expression had prognostic value for PFS and OS in patients with TNBC. Functional assays showed that *SEC62* expression stimulated TNBC cell migration. We conclude that *SEC62* is a migration-stimulating oncogene, a predictive marker of treatment response, and prognostic marker for survival in TNBC. These findings may have therapeutic implications, as they can be used for improving and tailoring treatment regimens for patients with TNBC in the future.

## Figures and Tables

**Figure 1 ijms-24-09576-f001:**
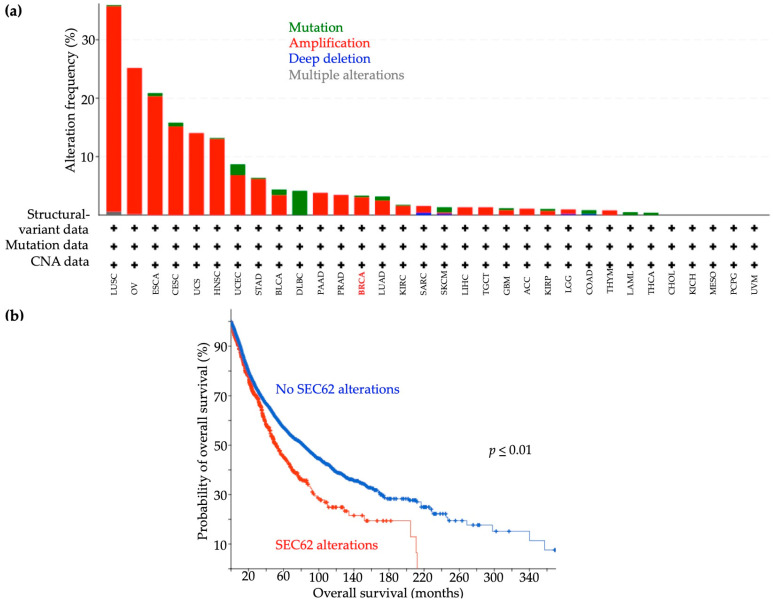
Type and frequency of reported SEC62 gene alterations and overall survival rates. (**a**) Type and frequency of SEC62 gene alterations recorded in the TCGA from the National Cancer Institute GDC Data Portal. The analysis was performed in 86,962 cases overall from 68 different primary tumor sites on 13 February 2023. CNA = copy number alteration, LUSC = lung squamous cell carcinoma, OV = ovarian serous cystadoncarcinoma, ESCA = esophageal adenocarcinoma, CESC = cervical squamous cell carcinoma, UCS = uterine carcinosarcoma, HNSC = head and neck squamous cell carcinoma, UCEC = uterine corpus endometrial carcinoma, STAD = stomach adenocarcinoma, BLCA = bladder urothelial carcinoma, DLBC = diffuse large B-cell lymphoma, PAAD = pancreatic adenocarcinoma, PRAD = prostate adenocarcinoma, BRCA = breast invasive carcinoma, LUAD = lung adenocarcinoma, KIRC= kidney renal clear cell carcinoma, SARC = sarcoma, SKCM = skin cutaneous melanoma, LIHC = liver hepatocellular carcinoma, TGCT = testicular germ cell tumors, GBM = glioblastoma multiforme, ACC = adrenocortical carcinoma, KIRP = kidney renal papillary cell carcinoma, LGG = brain lower grade glioma, COAD = colorectal adenocarcinoma, THYM = thymoma, LAML = acute myeloic leukemia, THCA = thyroid carcinoma, CHOL = cholangiocarcinoma, KICH = kidney chromophobe carcinoma, MESO = mesothelioma, PCPG = pheochromocytoma and paraganglioma, UVM = uveal melanoma. (**b**) Overall survival across all cancer entities recorded in the TCGA correlated with SEC62 alterations. The analysis was performed on 13 February 2023. Two-sided *p* values are indicated (log-rank test).

**Figure 2 ijms-24-09576-f002:**
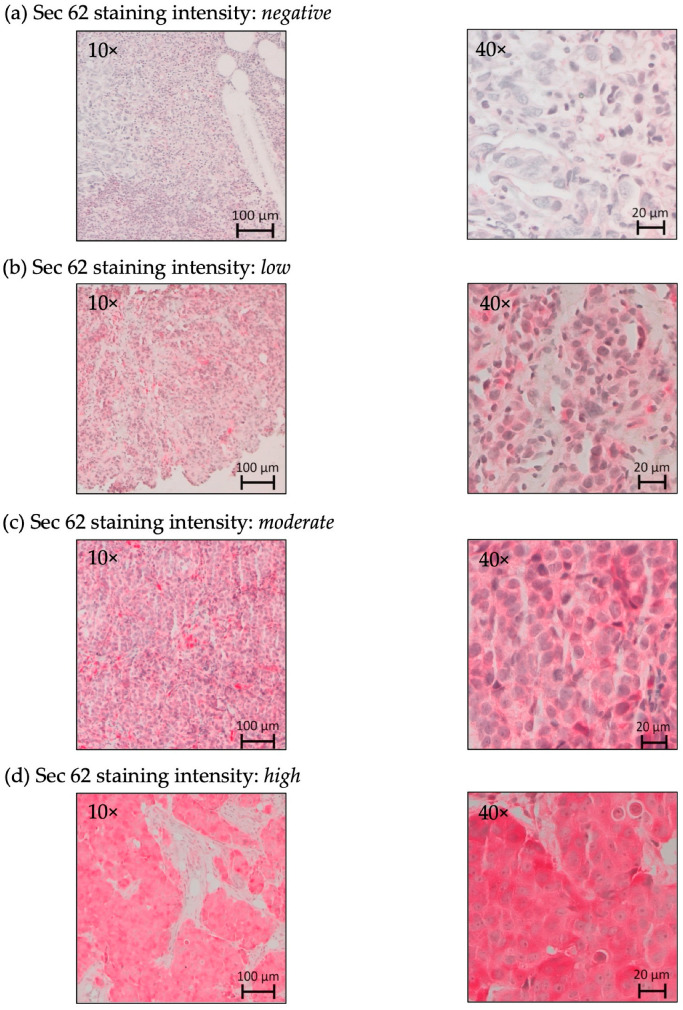
Sec62 immunohistochemistry (IHC) in triple-negative breast cancer: (**a**) negative SEC62 expression (immunoreactive score IRS 0–2) in normal breast tissue, (**b**) low (IRS 3–4), (**c**) moderate (IRS 6–8), (**d**) high (IRS 9–12) immunostaining intensity in triple-negative breast cancer. SEC62 expression is indicated by a red cytoplasmic signal; counterstaining was performed with hematoxylin (blue).

**Figure 3 ijms-24-09576-f003:**
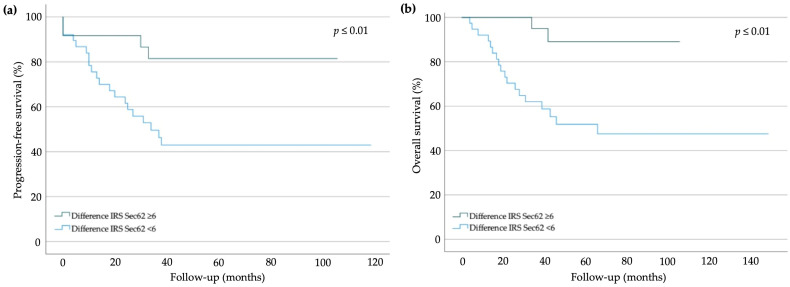
Survival rates for patients with primary TNBC after neoadjuvant chemotherapy: (**a**) progression-free and (**b**) overall survival in the whole cohort; Sec62 immunoreactive score (IRS) ≥ 6, Sec62 IRS < 6 (log-rank test).

**Figure 4 ijms-24-09576-f004:**
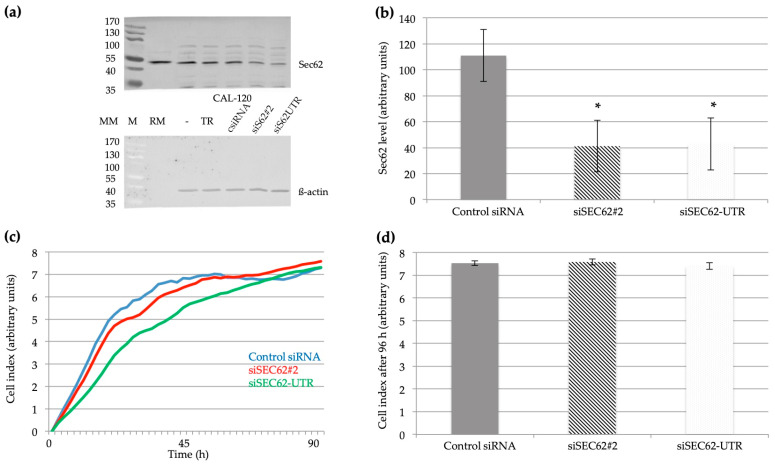
Effect of *SEC62* expression level on the proliferation of CAL-120 cells. (**a**,**b**) The Sec62 protein level was determined by Western blot using β-actin as a housekeeping protein. M, marker; MM, molecular mass in kDa; RM, canine pancreatic rough microsomes for Sec62 identification; TR, transfection reagent; * statistically significant compared with control siRNA. (**c**,**d**) Proliferation assay: unitless cell index was used as an indicator for cell proliferation of CAL-120 cells transfected with control siRNA, SEC62#2 siRNA, or SEC62-UTR siRNA as indicated. * Statistically significant compared with control siRNA. The results represent the median values of four (**b**) (*p^1^* ≤ 0.01; *p^2^* ≤ 0.01) and three (**d**) (*p^1^* = 0.18; *p^2^* = 0.18) independent experiments, respectively, and are shown with the standard error of the mean (sem).

**Figure 5 ijms-24-09576-f005:**
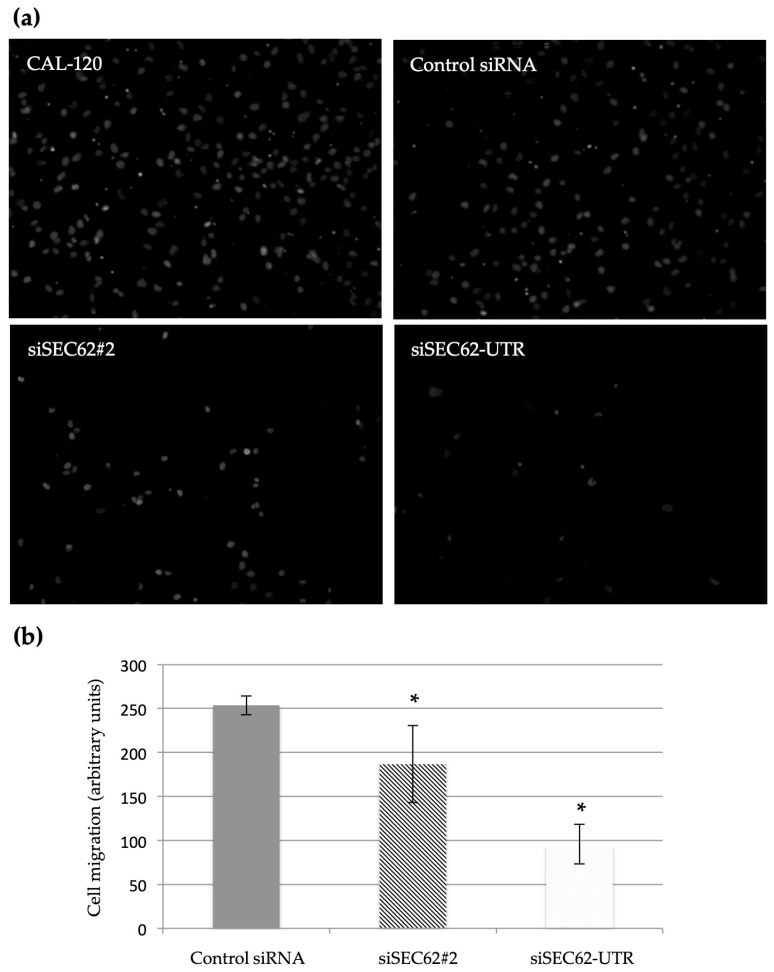
Effect of *SEC62* expression on the migration of CAL-120 cells analyzed in a Boyden chamber trans-well migration assay. The cells that migrated through the 8 µm sized pores of the insert system were fixed and labeled with DAPI (white dots). (**a**) Migrated CAL-120 cells either untreated, transfected with control siRNA, transfected with Sec62-siRNA-#2, or transfected with Sec62-siRNA-UTR (directed against the 3′ untranslated region of *SEC62*) as indicated using a ten-fold objective magnification. (**b**) Statistical analysis of the total number of migrated cells per chamber after 48 h (*p^1^* ≤ 0.01; *p^2^* ≤ 0.01). * Statistically significant compared with control siRNA. The quantitative results represent the median values of three independent experiments and are shown with the standard error of the mean (sem).

**Figure 6 ijms-24-09576-f006:**
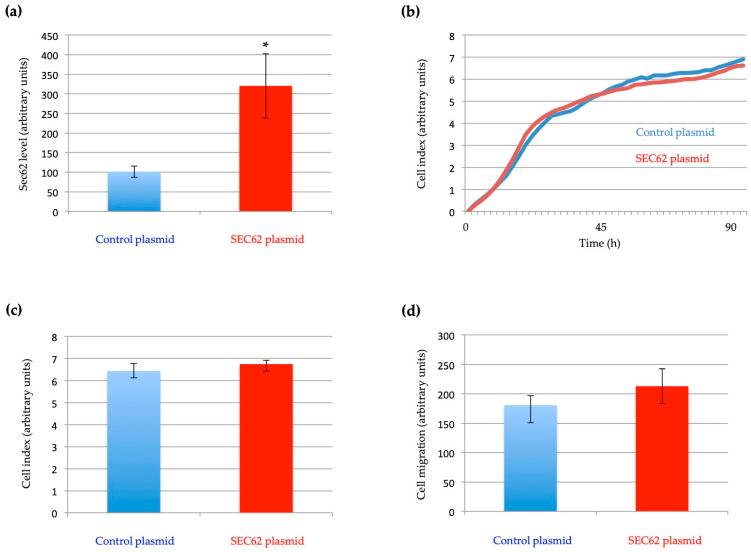
Effect of further *SEC62* overexpression on CAL-120 cells. Further *SEC62* overexpression in CAL-120 cells was accomplished via transfection with a *SEC62*-encoding expression plasmid (labeled in red). (**a**) The Sec62 protein level was determined by Western blot using β-actin as a housekeeping protein as indicated in Figure 4a,b (*p* ≤ 0.01). * Statistically significant compared with control plasmid. (**b**,**c**) Proliferation assay as described in Figure 4c,d: unitless cell index was used as an indicator for cell proliferation (*p* = 0.29). (**d**) Effect on the migration of CAL-120 cells was analyzed in a Boyden chamber trans-well migration assay as illustrated in Figure 5a,b. The indicated cells that migrated through the 8 µm sized pores of the insert system were fixed after 48 h and labeled with DAPI (*p* = 0.11). (**a**–**d**) The results represent the median values of three independent experiments, respectively, and are shown with the standard error of the mean (sem).

**Table 1 ijms-24-09576-t001:** Patients’ and tumor characteristics for patients with triple-negative breast cancer (TNBC) (*n* = 64). Regression grade according to Sinn [20].

	TNBC (*n* = 64)
	*n* (%)
**Age** (median (range))	55 (31–78)
**Menopausal status**	
Premenopausal	21 (32.8%)
Postmenopausal	43 (67.2%)
**cT-stage**	
T1	22 (34.4%)
T2	30 (46.9%)
T3	2 (3.1%)
T4	10 (15.6%)
**ypT-stage**	
T0	27 (42.2%)
T1	24 (37.5%)
T2	6 (9.4%)
T3	5 (7.8%)
T4	2 (3.1%)
**cN-stage**	
N0	31 (48.4%)
N1	20 (31.3%)
N2	6 (9.4%)
N3	7 (10.9%)
**ypN-stage**	
N0	44 (68.8%)
N1	10 (15.6%)
N2	7 (10.9%)
N3	3 (4.7%)
**Grading**	
G1	0 (0%)
G2	12 (18.7%)
G3	52 (81.3%)
**Ki67**	
<15%	0 (0%)
≥15%	64 (100%)
**Regression grade according to Sinn** (median (range))	2 (0–4)
**Local recurrence**	
Yes	4 (6.6%)
No	60 (93.4%)
**Distant metastasis**	
Yes	25 (39.1%)
No	39 (60.9%)

**Table 2 ijms-24-09576-t002:** Summary of immunohistochemical *SEC62*—expression analysis (*n* = 64).

	TNBC (*n* = 64)	*p*
	Median (Range)	
**IRS core biopsy**	8 (2–12)	
**IRS final specimen**	4 (0–12)	≤0.01
**Difference between IRS core biopsy and IRS final specimen**	4 (−6–12)	
	***n* (%)**	
**IRS core biopsy**		
negative	3 (4.7%)	
low	8 (12.5%)	
moderate	32 (50%)	
high	21 (32.8%)	
**IRS final specimen**		
negative	28 (44.4%)	
low	11 (17.5%)	
moderate	15 (23.8%)	
high	9 (14.3%)	
**IRS ≤ 8 core biopsy**	42 (65.6%)	
**IRS > 8 core biopsy**	22 (34.4%)	
**IRS ≤ 8 final specimen**	51 (81%)	
**IRS > 8 final specimen**	12 (19%)	
**Difference between IRS core biopsy and IRS final specimen < 6**	39 (61.9%)	
**Difference between IRS core biopsyand IRS final specimen ≥ 6**	24 (38.1%)	

**Table 3 ijms-24-09576-t003:** Correlation between Sec62 IRS core biopsy/final specimen and response to treatment(measured by regression grade according to Sinn et al.) (*n* = 64) [20].

	**IRS ≤ 8 Core Biopsy** ***n* = 42**	**IRS > 8 Core Biopsy** ***n* = 22**	** *p* **
**Regression grade according to Sinn** (median, range)	2 (0–4)	3 (1–4)	≤0.01
	**IRS ≤ 8 Final Specimen** ***n* = 51**	**IRS > 8 Final Specimen** ***n* = 12**	
**Regression grade according to Sinn** (median, range)	4 (0–4)	1 (0–2)	≤0.01
	**Difference between IRS Core Biopsy and IRS Final Specimen < 6** ***n* = 39**	**Difference between IRS Core Biopsy and IRS Final Specimen ≥ 6** ***n* = 24**	
**Regression grade according to Sinn** (median, range)	1 (0–4)	4 (2–4)	≤0.01

**Table 4 ijms-24-09576-t004:** Factors affecting regression grade according to Sinn et al. on linear regression analysis (ANOVA), 95% CI = confidence interval, in bold type statistically significant values (*n* = 64) [20].

	Odds Ratio	95% CI	*p*
**Age**	0.01	−0.02–0.02	0.94
**cT-stage**	−0.35	−0.66–−0.04	**0.03**
**cN-stage**	0.04	−0.25–0.32	0.78
**G**	0.36	−0.28–0.99	0.26
**Ki67**	0.01	−0.01–0.02	0.85
**IRS > 8 core biopsy**	0.41	−0.15–0.96	0.15
**IRS > 8 final specimen**	−0.51	−1.28–0.26	0.19
**Difference between IRS core biopsy and IRS final specimen ≥ 6**	1.75	1.16–2.34	**≤0.01**

## Data Availability

The dataset used and analyzed during the current study is available from the corresponding author upon reasonable request.

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
