# Peer review of "The 3q Oncogene SEC62 Predicts Response to Neoadjuvant Chemotherapy and Regulates Tumor Cell Migration in Triple Negative Breast Cancer"

_ijms, 2023, doi:10.3390/ijms24119576_

Round 1
Reviewer 1 Report
In manuscript IJMS-2399298, Radosa et al. propose a possible triple negative breast cancer driver function of the SEC62 gene which is located within a frequently amplified region on 3q26. They also present evidence for a role of SEC62 in cancer cell migration. The involvement of SEC62 in triple negative breast cancer appears plausible to me, however, the study falls short in convincing substantiation of its effect on cell proliferation and migration as the presented functional study is only preliminary. Additional experimentation would allow more solid conclusions.
Major points:
According to CRISPR screens, SEC62 is highly essential for CAL-120 cell proliferation and/or survival (https://depmap.org/portal/gene/SEC62?tab=dependency). Also in the study at hand, RNAi seemed to reduce proliferation (Fig. 4c) slightly but only temporarily. There was no significant effect on cell numbers after 96h (Fig. 4d) presumably due to dilution/degradation of anyway only moderately efficient siRNAs (protein reduction approx. 60%). Thus, since proliferation is severely affected by SEC62 knock-out, please rephrase lines 213-216 and 292-294. Furthermore, the observed decrease in cell migration (Fig. 5) may as well be an indirect effect due to reduced cellular fitness. Scratch or similar assays could provide valuable additional information.
I highly recommend additional functional experiments with other breast cancer cell lines (e.g. 3q amplified versus not amplified; CAL-51 or HCC1428 which are not SEC62 dependent according to depmap; or triple negative versus hormone responsive). Moreover, CRISPR/Cas9 mediated knock-out could be used to reduce SEC62 levels more efficiently. This would allow more robust conclusions about SEC62 function at least in a couple of breast cancer cell lines in vitro.
Minor points:
Line 150-152: reformulate: “All TNBC tissue slides analyzed showed SEC62 overexpression compared with that in surrounding tumor-free tissue. We observed cytoplasmatic Sec62 positivity in all breast cancer cells and no physiological breast tissue cells (figure 2).”
Line 437: protein band intensities were probably not quantified in kDa but in arbitrary signal units
Typos: e.g. lines 156 (immunoreactiescore), 163/4 [median … 4 (range, 6-12)]
Figure 4a: please explain rough microsome fractionation in more detail.
Also show results for SEC62 overexpression experiment even if there was no significant effect on proliferation or migration. In general, a statistical test (e.g. one-way ANOVA) should be performed and significance levels shown in 4b, 4d, 5b.
Please, generally describe the used reagents in more detail e.g. provide a RRID (Research Resource Identification) for antibodies especially for the self-made one.
Also disclose relevant information on the cell line CAL-120 in the text e.g. whether it is 3q amplified / SEC62 overexpressing, FGFR1 dependent etc. (Turner et al. Cancer Res 2010; Sharpe et al. Clin Cancer Res 2011).
A few typos and odd wordings must be corrected.
Author Response
Dear MS Xiong, dear reviewers,
we kindly thank you and your team for reviewing our manuscript entitled „The 3q oncogene SEC62 predicts response to neoadjuvant chemotherapy and regulates tumor cell migration in triple negative breast cancer“ and for giving us the chance to improve our manuscript with the revisions suggested by the reviewers. We kindly thank you for the thoughtful remarks, which clearly helped improving the manuscript and substantially enhanced our work. We have added a track-changed version of the manuscript to make the changes identifiable. Listed below are the point-to-point responses to the reviewers’ comments:
Reviewer 1
In manuscript IJMS-2399298, Radosa et al. propose a possible triple negative breast cancer driver function of the SEC62 gene which is located within a frequently amplified region on 3q26. They also present evidence for a role of SEC62 in cancer cell migration. The involvement of SEC62 in triple negative breast cancer appears plausible to me, however, the study falls short in convincing substantiation of its effect on cell proliferation and migration as the presented functional study is only preliminary. Additional experimentation would allow more solid conclusions.
Comment
According to CRISPR screens, SEC62 is highly essential for CAL-120 cell proliferation and/or survival (https://depmap.org/portal/gene/SEC62?tab=dependency). Also in the study at hand, RNAi seemed to reduce proliferation (Fig. 4c) slightly but only temporarily. There was no significant effect on cell numbers after 96h (Fig. 4d) presumably due to dilution/degradation of anyway only moderately efficient siRNAs (protein reduction approx. 60%). Thus, since proliferation is severely affected by SEC62 knock-out, please rephrase lines 213-216 and 292-294. Furthermore, the observed decrease in cell migration (Fig. 5) may as well be an indirect effect due to reduced cellular fitness. Scratch or similar assays could provide valuable additional information.
Response
Thank you very much for kindly reviewing our manuscript and for your helpful comments. We felt that incorporating your remarks and advice for more precise definitions and re-interpretation of functional assays in our manuscript substantially enhanced our work. We made amendments and added respective sections to the manuscript according to your suggestion. We incorporated a clearer interpretation of the functional assays in the result section, rephrased lines 213-216 and added the essential impact of SEC62 on CAL-120 cell proliferation and survival as a potential bias in the discussion section.
We strongly agree with your statement that the observed decrease in cell migration might partly be an indirect effect of reduced cellular fitness and added a respective statement into the discussion section. We also shared your thoughts on the possible reduced proliferation rate resulting in a reduced migration in a scratch assay and did not incorporate this approach therefore in the first place but additional techniques such as scratch or similar assays can be performed, but would take an additional time and would have made it impossible to match the revision deadline. We rephrased lines 292-294 as recommended and included your valuable input in the limitations section of the discussion.
“After the transfection of CAL-120 cells with two different SEC62-targeting siRNAs, median Sec62 protein levels were reduced to 41.25% for the SEC62#2 siRNA and to 42.75% for the SEC62-UTR siRNA (p ≤ 0.01) (figure 4 a, b), which did not affect TNBC cell proliferation after 96 hours (figure 4 c, d). We observed a temporary, but not statistically significant effect of SEC62-silencing on cell proliferation after 45 hours with median cell indices of 6.78 (range 6.70-7.15) for the Control siRNA, 6.21 (range 6.01-6.85) for the SEC62#2 siRNA and 5.08 (range 5.03-5.64) for the SEC62-UTR siRNA (p = 0.05) (figure 4 c). Median cell indices after 96 hours were 7.5 (range 7.4-7.6) for the Control siRNA, 7.6 (range 7.5-7.7) for the SEC62#2 siRNA and 7.4 (range 7.3-7.5) for the SEC62-UTR siRNA (p = 0.37) (figure 4 c, d). In the migration assays, SEC62 knock-down significantly reduced the cells’ migratory potential, with median cell migration arbitrary units of 263 (range 241-279) for the Control siRNA, 178 (range 96-276) for the siSEC62#2 and 72 (range 70-149) for the siSEC62-UTR (p ≤ 0.01) (figure 5 a, b).”
“A possible bias in the observed effect of SEC62 silencing on cell migration might be the SEC62 mediated temporary effect on cell proliferation and hence an indirect effect due to reduced cellular fitness. These findings are supported by CRISPR screens analyses (https://depmap.org/portal/gene/SEC62?tab=dependency), which demonstrate SEC62 to be relevant for CAL-120 cell proliferation and survival outcomes. However, the strong decrease in cell migration in contrast to the temporary effect on cell proliferation, hints at an independent effect of SEC62 on cell migration.”
Comment
I highly recommend additional functional experiments with other breast cancer cell lines (e.g. 3q amplified versus not amplified; CAL-51 or HCC1428 which are not SEC62 dependent according to depmap; or triple negative versus hormone responsive).
Response
We included more details on the functional experiments with other breast cancer cell lines which were performed by us as part of this project. Those were hormone receptor positive cells: ZR-75-30 (ER positive, HER2 positive, SEC62 gene effect_RNAi -0.04 Z score: 0.39; the growth patterns of this cell line did not allow us to interpret the migration assays) and BT474 (ER positive, HER2 positive, SEC62 gene effect_RNAi -0.17 Z score: -0.36; this cell line was not transfectable), Her2 positive cells: AU565 (ER negative, HER2/neu positive, SEC62 gene effect_CRISPR -0.49 Z score: -0.85, SEC62 gene effect_ RNAi -0.13 Z score: -0.12; we are currently working on this cell line and can not comment on them yet) and CAL120 (ER negative, HER2 negative, SEC62 gene effect_CRISPR -1.05 Z score: -3.46, SEC62 gene effect_ RNAi -0.01 Z score: 0.06) and other triple negative breast cancer cells: MDA-MB-453 (ER negative, HER2 negative, SEC62 gene effect_CRISPR -0.26 Z score: 0.26, SEC62 gene effect_ RNAi 0.11 Z score: 1.34; this cell line was not transfectable) and MDA-MB-231(ER negative, HER2 negative, SEC62 gene effect_CRISPR -0.37 Z score: 0.27, SEC62 gene effect_ RNAi -0.06 Z score: 0.31; we could not see an effect on the functional assays).
Comment
Moreover, CRISPR/Cas9 mediated knock-out could be used to reduce SEC62 levels more efficiently. This would allow more robust conclusions about SEC62 function at least in a couple of breast cancer cell lines in vitro.
Response
Thank you for this valuable remark. We used two different SEC62 targeting siRNAs in the first place to examine the effect of SEC62 on migration and proliferation on triple negative breast cancer cells lines as these siRNAs were previously established for various other cell lines.
However, we strongly agree with your statement that additional approaches would be helpful in order to further manifest the observations seen in the functional assays. In detail a CRISPR/Cas9 mediated knock-out could be used to reduce SEC62 levels. In cooperation with our corresponding working group, who developed a CRISPR/Cas9 mediated knock-out in HNSCC cells (Körner S, Pick T, Bochen F, Wemmert S, Körbel C, Menger MD, Cavalié A, Kühn JP, Schick B, Linxweiler M. Antagonizing Sec62 function in intracellular Ca2+ homeostasis represents a novel therapeutic strategy for head and neck cancer. Front Physiol. 2022 Aug 15;13:880004. doi: 10.3389/fphys.2022.880004), we are currently working on such an additional approach, but the development and establishment of these knock-out cell lines are scheduled to take until the end 12/2024 and could therefore not be included in this special issue. We rephrased lines 213-216 and have added an explanation why we used two different SEC62 targeting siRNAs and transient Sec62 depletion under our established conditions rather than knock out cells to the Material and Methods section: “We compared the migratory potential of CAL-120 cells, which were transfected with control siRNA or one of two different SEC62-targeting siRNAs in the BD Falcon FluoroBlok system. In addition, proliferation rates of the siRNA-treated cells were ana-lyzed in real-time in the xCELLigence SP system. These siRNAs were previously es-tablished for various other cell types and shown to cause efficient transient Sec62 depletion with little effect on cell proliferation during 96 hours of cell growth [17]. Because of this latter observation by different laboratories, and as we wanted to be able to compare our results with those found for other tumor cells, we applied transient knock-down with siRNAs rather than CRISPR/Cas9 mediated knock-out strategies.”
Comment
Line 150-152: reformulate: “All TNBC tissue slides analyzed showed SEC62 overexpression compared with that in surrounding tumor-free tissue. We observed cytoplasmatic Sec62 positivity in all breast cancer cells and no physiological breast tissue cells (figure 2).”
Response
Thank you for this amendment, we rephrased the sentence as suggested: “In the analyzed tissue slides, we observed cytoplasmic Sec62 positivity in all TNBC cells and not in physiological breast tissue cells (figure 2).”
Comment
Line 437: protein band intensities were probably not quantified in kDa but in arbitrary signal units
Response
Thank you, the sentence was rephrased as suggested. “The Sec62 and β-actin levels were quantified (in arbitrary signal units), and the former was normalized against the latter.”
Comment
Typos: e.g. lines 156 (immunoreactiescore), 163/4 [median … 4 (range, 6-12)]
Response
We sincerely apologize for these mistakes and corrected all the typos.
Comment
Figure 4a: please explain rough microsome fractionation in more detail.
Response
The details were added to the legend of figure 4 and Materials and Methods as suggested: “Canine pancreatic rough microsomes (RM) were analyzed in parallel and allowed the identification of the Sec62 protein in the human cell extracts [13, 14, 17, 24].”
Comment
Also show results for SEC62 overexpression experiment even if there was no significant effect on proliferation or migration. In general, a statistical test (e.g. one-way ANOVA) should be performed and significance levels shown in 4b, 4d, 5b.
Response
Please excuse this. The results are shown in new Figure 6 as suggested and described in the results section. “The induction of further SEC62 overexpression in CAL-120 cells via transfection with a SEC62-encoding expression plasmid did not alter cell proliferation or increase cell migration as compared to control plasmid-transfected cells (figure 6a-d). The median cell indices were 6.77 (range 5.66-6.9) for the Control plasmid and 6.62 (range 6.46-7.16) for the SEC62-encoding expression plasmid (p = 0.29). The median migrated cells in the Control plasmid group were 162 (range 161-220) and 182 (range 173-285) for the SEC62-encoding expression plasmid (p = 0.11). We attribute this to the observation that there already is a high level of SEC62 overexpression associated with TNBC (figure 2). Notably, in previous experiments SEC62 overexpression stimulated the migratory potential and stress tolerance of otherwise poorly migrating cells such as HEK293, HeLa and FaDu cells [13, 17].”
Thank you for the remark concerning the statistical tests, we added them in the results section.
Comment
Please, generally describe the used reagents in more detail e.g. provide a RRID (Research Resource Identification) for antibodies especially for the self-made one.
Response
The reagents were described in more detail in Materials and Methods as suggested: “The house-made rabbit antibody was directed against the COOH terminal undecapeptide of the human Sec62 protein and an aminoterminal cysteine (peptide sequence in single letter code: CGETPKSSHEKS)…..…We used the above described affinity-purified polyclonal rabbit antipeptide antibody directed against the C terminus of human Sec62 (at a dilution of 1:500) and a monoclonal mouse antibody directed against the N terminus of human ß-actin (at a dilution of 1:10,000) (Sigma-Aldrich, St. Louis, MO, USA). The secondary antibodies used were ECL Plex goat anti-rabbit Cy5 and anti-mouse Cy3 conjugates (GE Healthcare, Munich, Germany) (at dilutions of 1:1,000 and 1:10,000, respectively).”
However, we refrained from adding our antibody to PRID because we cannot provide the antibody for routine clinical purposes (although we routinely send it out for research purposes upon request), but added a statement to the IHC section of Materials and Methods that reads: “Notably, commercial anti-Sec62 antibodies were described as suitable alternative [23].”
Comment
Also disclose relevant information on the cell line CAL-120 in the text e.g. whether it is 3q amplified / SEC62 overexpressing, FGFR1 dependent etc. (Turner et al. Cancer Res 2010; Sharpe et al. Clin Cancer Res 2011).
Response
The cell line was described in more detail in Materials and Methods as suggested. “CAL-120 cells, derived from a patient with invasive TNBC, were obtained from the German Collection of Microorganisms and Cell Cultures GmbH/DSMZ (ACC 459).” CAL120 are FGFR1 overexpressing (Turner et al. Cancer Res 2010).

Reviewer 2 Report
In this study, authors have investigated SEC62 expression in pre- and post-NACT tissue samples from patients with TNBC. They conclude that SEC62 is overexpressed in TNBC and serves as a predictive marker for the response to NACT, a prognostic marker for PFS and OS, and a migration-stimulating oncogene in TNBC.
Previously, same authors had found that Sec62 protein expression was greater in breast-cancer tissue samples than in tumor-free tissue samples from the same patients. In addition, they had found that Sec62 expression was the highest in TNBC patients.
In my opinion, this work fits the scopus of International Journal of Molecular Sciences. However, it is more translational-clinical in nature than average for papers published in IJMS. Therefore, I strongly suggest to further extend the section dedicated to molecular and cellular experiments.
In addition, this work has some limitations, and several changes should be performed prior publication:
- You should describe what is the specific combination of NACT in your cohort of patients. Maybe your results are dependent on the type of neoadjuvant therapy that is administered (i.e. SEC62 expression is a predictive marker for the response to NACT with XXXX). Please describe it and include your respective thoughts in discussion section.
- Line 28: please include the number of samples.
- Line 32: please better define “oncological outcomes” for readers who are not in the same field.
- Line 162: you should define core biopsy tissue specimens and final specimens. According to MYM section, I suppose that core biopsy tissue specimens refer to prior NACT and final specimens refer to after NACT, but please explain it when the results are described.
- Figure 3a-3b: p ≤ 0.01 is included in the figure caption, but in the text (lines 189-204) is described that there are no significant differences. Please revise and better describe these results. In addition, in lines 259-260 it is stated that “IRS differences < 6 were associated with shortened 259 PFS and OS for patients with TNBC”.
- Line 213-215: You should extend this section. Please differentiate Figure 4a, b, c, d in the text. Include more detailed information.
- Line 215-216: You should extend this section. Please differentiate Figure 5a, b in the text. Include more detailed information.
- Line 215-216: are these differences significant?
- Figures 4 and 5: statistical analyses are missing, please include it.
- Line 454: Please complete the following sentence: “SEC62 expression was increased in…”.
Author Response
Dear MS Xiong, dear reviewers,
we kindly thank you and your team for reviewing our manuscript entitled „The 3q oncogene SEC62 predicts response to neoadjuvant chemotherapy and regulates tumor cell migration in triple negative breast cancer“ and for giving us the chance to improve our manuscript with the revisions suggested by the reviewers. We kindly thank you for the thoughtful remarks, which clearly helped improving the manuscript and substantially enhanced our work. We have added a track-changed version of the manuscript to make the changes identifiable. Listed below are the point-to-point responses to the reviewers’ comments:
Reviewer 2
Comment
In this study, authors have investigated SEC62 expression in pre- and post-NACT tissue samples from patients with TNBC. They conclude that SEC62 is overexpressed in TNBC and serves as a predictive marker for the response to NACT, a prognostic marker for PFS and OS, and a migration-stimulating oncogene in TNBC. Previously, same authors had found that Sec62 protein expression was greater in breast-cancer tissue samples than in tumor-free tissue samples from the same patients. In addition, they had found that Sec62 expression was the highest in TNBC patients.
In my opinion, this work fits the scopus of International Journal of Molecular Sciences. However, it is more translational-clinical in nature than average for papers published in IJMS. Therefore, I strongly suggest to further extend the section dedicated to molecular and cellular experiments.
Response
Thank you very much for this valuable advice.
Thank you very much for kindly reviewing our manuscript and for your helpful comments. We felt that incorporating your remarks and additional analyses in our manuscript substantially enhanced our work. We made amendments and added respective sections to the manuscript according to your suggestion. We agree with you statement that the work presented is more translational-clinical in nature than the average papers published in IJMS and apologize that we have not indicated during the submission process that the manuscript was invited for the „Special Issue on Advances in Molecular Research of Oncogenes“. The purpose of this Special Issue is „to explore the expanding field of molecular research of Oncogenes to provide valuable translational clues that can help in anticipating relevant clinical needs“. Therefore, the manuscript is very focused on the translational aspect of our findings. We have revised the manuscript and extended the section on molecular and cellular experiments to better fulfill this purpose.
Comment
In addition, this work has some limitations, and several changes should be performed prior publication: You should describe what is the specific combination of NACT in your cohort of patients. Maybe your results are dependent on the type of neoadjuvant therapy that is administered (i.e. SEC62 expression is a predictive marker for the response to NACT with XXXX). Please describe it and include your respective thoughts in discussion section.
Response
Thank you for this valuable comment, we added the combination of NACT to the text (4x Epirubicin 90 mg/m2 + Cyclophosphamid 600 mg/m2 i.v. q2w, followed by 12x Paclitaxel 80 mg/m2 + Carboplatin AUC 2 i.v. q1w) and also included respective thoughts in the discussion and limitations section.
Comment
Line 28: please include the number of samples.
Response
The number was added as suggested: “64”
Comment
Line 32: please better define “oncological outcomes” for readers who are not in the same field.
Response
Thank you, the definition was given as suggested: “progression free and overall survival.”
Comment
Line 162: you should define core biopsy tissue specimens and final specimens. According to MYM section, I suppose that core biopsy tissue specimens refer to prior NACT and final specimens refer to after NACT, but please explain it when the results are described.
Response
We redefined core biopsy tissue specimens and final specimens as suggested in the results’ section: “prior to NACT…post NACT”
Comment
Figure 3a-3b: p ≤ 0.01 is included in the figure caption, but in the text (lines 189-204) is described that there are no significant differences. Please revise and better describe these results. In addition, in lines 259-260 it is stated that “IRS differences < 6 were associated with shortened 259 PFS and OS for patients with TNBC”.
Response
Thank you for this comment, we revised these results to better emphasize the statistically significant effect of IRS differences < 6 on PFS and OS: “ When analyzing correlations between SEC62 expression and progression free survival (PFS), no correlation between the mean PFS and an IRS cutoff of 8 on core biopsy (77.8 months (95% CI 62.0-93.6), 61.9% vs. 60.1 months (95% CI 40.5-79.7), 60.0%; p = 0.69) or between PFS and an IRS cutoff of 8 on final specimen (78.1 months (95% CI 64.0-92.2), 63.0% vs. 50.2 months (95% CI 19.7-82.6), 44.4%; p = 0.23) was detected. But, we observed a mean PFS of 89.5 (95% CI 74.9-104.2) months (83.3%) in the group of patients with a difference of the Sec62 IRS ≥ 6 between pre- and post-treatment, compared to 60.7 (95% CI 43.5-77.9) months (47.4%) in the group of patients with a difference of the Sec62 IRS < 6 (p ≤ 0.01) (figure 3a). For overall survival (OS), an IRS cutoff of 8 on core biopsy (82.7 months (95% CI 68.9-97.4), 64.3% vs. 108.0 months (95% CI 78.7-137.3), 75.0%; p = 0.51) or on final specimen (106.1 months (95% CI 88.9-123.4), 70.4% vs. 56.1 months (95% CI 27.7-84.6), 44.4%; p = 0.08) did not correlate with the mean OS. Though, we detected a mean OS of 98.6 (95% CI 88.9-100.3) months (91.7%) in the group of patients with a difference of the Sec62 IRS ≥ 6 between pre- and post-treatment and a mean OS of 84.6 (95% CI 63.5-105.7) months (52.6%) in the group of patients with a difference of the Sec62 IRS < 6 between pre- and post-treatment (p ≤ 0.01) (figure 3b).”
Comment
Line 213-215: You should extend this section. Please differentiate Figure 4a, b, c, d in the text. Include more detailed information.
Response
Thank you for your comment, this section was extended according to your suggestion. “After the transfection of CAL-120 cells with two different SEC62-targeting siRNAs, median Sec62 protein levels were reduced to 41.25% for the SEC62#2 siRNA and to 42.75% for the SEC62-UTR siRNA (p ≤ 0.01) (figure 4 a, b), which did not affect TNBC cell proliferation after 96 hours (figure 4 c, d). We observed a temporary, but not statistically significant effect of SEC62-silencing on cell proliferation after 45 hours with median cell indices of 6.78 (range 6.70-7.15) for the Control siRNA, 6.21 (range 6.01-6.85) for the SEC62#2 siRNA and 5.08 (range 5.03-5.64) for the SEC62-UTR siRNA (p = 0.05) (figure 4 c). Median cell indices after 96 hours were 7.5 (range 7.4-7.6) for the Control siRNA, 7.6 (range 7.5-7.7) for the SEC62#2 siRNA and 7.4 (range 7.3-7.5) for the SEC62-UTR siRNA (p = 0.37) (figure 4 c, d). In the migration assays, SEC62 knock-down significantly reduced the cells’ migratory potential, with median cell migration arbitrary units of 263 (range 241-279) for the Control siRNA, 178 (range 96-276) for the siSEC62#2 and 72 (range 70-149) for the siSEC62-UTR (p ≤ 0.01) (figure 5 a, b).”
Comment
Line 215-216: You should extend this section. Please differentiate Figure 5a, b in the text. Include more detailed information.
Response
Thank you for this remark, the section was extended as suggested: ”The induction of further SEC62 overexpression in CAL-120 cells via transfection with a SEC62-encoding expression plasmid did not alter cell proliferation or increase cell migration as compared to control plasmid-transfected cells (figure 6a-d). The median cell indices were 6.77 (range 5.66-6.9) for the Control plasmid and 6.62 (range 6.46-7.16) for the SEC62-encoding expression plasmid (p = 0.29). The median migrated cells in the Control plasmid group were 162 (range 161-220) and 182 (range 173-285) for the SEC62-encoding expression plasmid (p = 0.11). We attribute this to the observation that there already is a high level of SEC62 overexpression associated with TNBC (figure 2). Notably, in previous experiments SEC62 overexpression stimulated the migratory potential and stress tolerance of otherwise poorly migrating cells such as HEK293, HeLa and FaDu cells [13, 17].“
Comment
Line 215-216: are these differences significant?
Response
We sincerely apologize for the omission of these values, we added the p values.
Comment
Figures 4 and 5: statistical analyses are missing, please include it.
Response
Please excuse these mistakes, we added the p values to the legends of the respective figures.
Comment
Line 454: Please complete the following sentence: “SEC62 expression was increased in…”
Response
We sincerely apologize for this mistake, which occurred due to formatting issues, we included the completed senctence in the manuscript. “This study showed that SEC62 expression was increased in TNBC and predicted the re-sponse to NACT with 4x Epirubicin 90 mg/m2 + Cyclophosphamid 600 mg/m2 i.v. q2w, followed by 12x Paclitaxel 80 mg/m2 + Carboplatin AUC 2 i.v. q1w, and that the dynamics of this expression had prognostic value for PFS and OS, in patients with TNBC. Functional assays showed that SEC62 expression stimulated TNBC cell migration.”

Round 2
Reviewer 1 Report
The authors responded satisfactorily to many of my and the other reviewer's points. The body of evidence for the role of SEC62 in proliferation and migration appears still weak to me. However, I agree with publication.
proofreading by a native speaker would be advised
Reviewer 2 Report
I consider that the current version of this manuscript could be accepted for publication in International Journal of Molecular Sciences Journal. The authors have successfully addressed all the comments. Consequently, the manuscript has been improved.